# Two-Year Results of 0.01% Atropine Eye Drops and 0.1% Loading Dose for Myopia Progression Reduction in Danish Children: A Placebo-Controlled, Randomized Clinical Trial

**DOI:** 10.3390/jpm14020175

**Published:** 2024-02-02

**Authors:** Niklas Cyril Hansen, Anders Hvid-Hansen, Flemming Møller, Toke Bek, Dorte Ancher Larsen, Nina Jacobsen, Line Kessel

**Affiliations:** 1Department of Ophthalmology, Copenhagen University Hospital—Rigshospitalet-Glostrup, DK-2600 Glostrup, Denmark; anders.johan.hvid-hansen@regionh.dk (A.H.-H.); njac@dadlnet.dk (N.J.); line.kessel.01@regionh.dk (L.K.); 2Department of Ophthalmology, University Hospital of Southern Denmark—Vejle Hospital, DK-7100 Vejle, Denmark; flemming.moller2@rsyd.dk; 3Department of Ophthalmology, Aarhus University Hospital, DK-8200 Aarhus N, Denmark; toke.bek@mail.tele.dk (T.B.); dortlars@rm.dk (D.A.L.); 4Department of Clinical Medicine, University of Copenhagen, DK-2200 København N, Denmark

**Keywords:** myopia, myopia control, low-dose atropine, axial length, spherical equivalent

## Abstract

We investigated the two-year safety and efficacy of 0.1% loading dose and 0.01% low-dose atropine eye drops in Danish children for reduction in myopia progression in an investigator-initiated, placebo-controlled, double-masked, randomized clinical trial. Ninety-seven six- to twelve-year old myopic participants were randomized to 0.1% loading dose for six months and then 0.01% for eighteen months (loading dose group, N = 33), 0.01% for two years (0.01% group, N = 32) or placebo for two years (placebo, N = 32). Axial length (AL) and spherical equivalent refraction (SER) were primary outcomes. Secondary outcomes included adverse events and reactions, choroidal thickness, and other ocular biometrical measures. Outcomes were measured from baseline and at six-month intervals. Individual eyes nested by participant ID were analyzed with linear-mixed model analysis. Data were analyzed with intention-to-treat. Mean AL was 0.08 mm less (95% confidence interval (CI): −0.01; 0.17, *p*-value = 0.08) in the 0.1% loading dose and 0.10 mm less (95% CI: 0.01; 0.19, *p*-value = 0.02) in the 0.01% group after two years of treatment compared to placebo. Mean SER progression was 0.12 D (95% CI: −0.10; 0.33) less in the loading dose and 0.26 D (95% CI: 0.04; 0.48) less in the 0.01% groups after two years of treatment compared to placebo (*p*-value = 0.30 and 0.02, respectively). In total, 17 adverse events were reported in the second-year follow-up, and all were rated as mild. Adjusting for iris color did not affect treatment effect estimates. Intra-ocular pressure increased over two years comparably between all groups but remained within normal limits. Two-year treatment with 0.01% low-dose atropine eye drops is a safe and moderately efficacious intervention in Danish children for reducing myopia progression.

## 1. Introduction

Myopia prevalence has increased rapidly in recent years in many parts of the world and the increase is expected to continue [1]. While the prevalence of myopia in Denmark is lower and seems relatively stable [2], a recent study nevertheless found a myopia prevalence of 17.9% in Danish adolescents [3]. Higher degrees of myopia increase the risk of long-term, and potentially blinding, complications such as retinal detachment, myopic choroidal neovascularization and glaucoma [4]. This increased risk of myopia complications makes it imperative to find viable interventions to reduce myopia progression and thereby the prevalence of high myopia.

Many interventions have been investigated in an effort to delay and reduce myopia progression, including, among others, lifestyle changes (increased outdoor time and decreased amount of near-work) [5,6], optical interventions (multifocal segments spectacle lenses, orthokeratology and multifocal lenses) [7,8,9] and pharmacological interventions [10,11,12,13]. Contact lens use is associated with a risk of infectious keratitis that may result in permanent visual loss and should be used with caution in children [14].

Pharmacological interventions present an alternative due to their limited side effects and ease of administration. Low-dose atropine eye drop treatment is at the moment the most commonly prescribed pharmacological intervention [15], with a proven dose-dependent effect [10] on myopia progression in Asian children [16,17]. The Atropine for the Treatment Of Myopia (ATOM) 2 study found a superior effect and a lower number of side effects in their 0.01% atropine group compared to their 0.1% atropine group and speculated that this was the result of a lesser rebound effect in their 0.01% atropine group after treatment cessation [18]. Similar efficacy of 0.01% atropine was previously found in other studies in myopic Asian children [17,19]. The Low-concentration Atropine for Myopia Progression (LAMP) 2 study has since suggested that 0.05% is a better trade-off between efficacy and side effects in this population [20].

There could be differences in effects and side effects of low-dose atropine between ethnicities, as an example because of the lighter pigmentation of the iris in White children [21,22]. A few studies have examined the efficacy and safety of low-dose atropine in children outside Asia (including our previous analyses [23,24]), with conflicting results (Table 1) [24,25,26,27,28].

A North American study examining the efficacy of 0.01% atropine found a reduced axial length (AL) progression of 0.002 mm in their 0.01% group compared to placebo after two years of treatment, i.e., no discernible reductive effect on myopia progression, in an ethnically heterogenous sample of children with a mean age of 10.1 years [26]. Zadnik et al., 2023 found a statistically significant 0.12 mm reduction in axial elongation at two years in their 0.01% group compared to placebo in US children with mixed ethnicity (51.2% White) and a mean age of 8.9 years [27]. Most of the non-Asian studies have been carried out in ethnically heterogenous cohorts [26,27,28,29], in different environments, and therefore it might be difficult to discern the effect of environment vs. ethnicity. A recently published Irish study, The Myopia Outcome Study of Atropine in Children (MOSAIC) study, examined the effect of 0.01% atropine on AL in a cohort of predominantly White six- to sixteen-year-old children and found a statistically significant 0.07 mm reductive effect on axial elongation at two-year follow-up [25], with a larger and only statistically significant effect in White children [25]. Environment might also play a part. For example, the Western Australia-ATOM study found a non-significant 0.05 mm reduction in axial elongation at two-year follow-up [28]. Interestingly, when performing subgroup analysis based on ancestry, they found a statistically significant reductive effect on AL in children of European descent at all visits except the two-year follow-up visit, but in contrast to LAMP [30] they did not find a statistically significant reductive effect in Asian children, although they questioned whether their study was powered for this comparison [28]. Our previous six- and twelve-month interim analysis shows a comparable efficacy in Danish children to that observed in LAMP at one-year follow-up [23,24].

In this analysis, we investigated the safety and efficacy of a two-year intervention with low-dose atropine eye drops to attempt to reduce myopia progression in Danish children aged six to twelve years at inclusion. Additionally, we wanted to examine if the increased effect of an initial six-month 0.1% loading dose was sustained after dose-switching to 0.01%.

## 2. Materials and Methods

### 2.1. Study Design

The study was a placebo-controlled, double-masked, investigator-initiated, randomized clinical trial, investigating the efficacy and safety of low-dose atropine eye drops in myopic Danish children. Children between six and nine years of age with a spherical power of ≤−1 diopter (D) in at least one eye and between nine and twelve years old with ≤−2 D in at least one eye were included. The higher ≤−2 D criterion for the older age group was chosen to make sure all participants experienced myopia progression, since it was not possible to retrieve data on progression rates prior to study inclusion. Maximum astigmatism allowed at inclusion was less than −1.5 D. Exclusion criteria were children presenting with myopia secondary to retinal dystrophies, collagenopathies (in particular Ehlers–Danlos, Marfan and Sticklers syndromes), other ocular conditions, eye surgery before inclusion, use of other myopia control methods before inclusion, not being able to comply with eye examinations, serious systemic health issues or developmental disorders or delays. Participants were seen at baseline and for follow-up visits at three, six, nine, twelve, eighteen and twenty-four months. Examinations took place at three different research settings at the Departments of Ophthalmology at Aarhus University Hospital, University Hospital of Southern Denmark—Vejle Hospital, and Copenhagen University Hospital—Rigshospitalet Glostrup, geographically spread across Denmark. The study design has previously been thoroughly described in a twelve-month interim analysis [24].

### 2.2. Trial Registration and Ethics Approval

The study was posted at www.clinicaltrials.gov (NCT03911271), accessed on 11 April 2019, before the initiation of the trial and it was registered in the European Clinical Trials Database (EudraCT, number: 2018-001286-16). The trial received approval from the Committees on Health Research Ethics for the Capital Region of Denmark (reference number: H-18043987), the Danish Data Protection Agency (reference number: P-2022-85) and the Danish Medicines Agency (reference number: 2018040088). The study followed the tenets of the Declaration of Helsinki and all parents of study participants gave written informed parental consent. Study participants themselves verbally consented to participate in the study. The good clinical practice (GCP) units at Aarhus, Odense and Copenhagen University Hospitals provided quality assurance.

### 2.3. Interventions

Participants were randomized via algorithm 1:1:1 to either 0.01% atropine eye drops for two years (0.01% group) or 0.1% loading dose for six months followed by 0.01% for 18 months (0.1% loading dose group) or vehicle eye drops for the two-year intervention period (placebo). The intervention was applied nightly in both eyes. Compliance was evaluated using parent-administered checklists with boxes to note the daily use of the eye drops. Participants were compliant when they used the drops 75% of the time, in accordance with the compliance threshold outlined in the ATOM 2-study [11]. Photochromatic or near-add glasses were refunded in cases of relevant visual adverse events (photophobia or near vision difficulties). The two-year intervention was followed by a wash-out period of one-year.

### 2.4. Outcomes

Myopia progression was the primary outcome, measured by AL in non-cycloplegic eyes and spherical equivalent refraction (SER) in cycloplegic eyes. Changes from baseline to two years in macular choroidal thickness, keratometry, lens thickness, anterior chamber depth (ACD) and adverse events and reactions (AE/AR) were secondary outcome measures.

### 2.5. Power Calculation, Sample Size and Randomization Procedure

The study power was calculated using a previously reported SER progression in myopic Danish school children [31]. In order to detect a 50% reduction in progression after 36 months of treatment, with 80% power and a significance level of 0.05, a sample size in each intervention group of a minimum of 21 participants was needed. To account for an at the time unknown effect size of low-dose atropine in Danish children, potential drop-out and the length of the study, additional study participants were recruited. The randomization procedure has been thoroughly described in our previous analyses [23,24]. Allocation concealment was achieved by masking randomization status from both parents, participants and trial staff. Statistical analyses were also performed masked.

### 2.6. Examinations

All examinations were performed at each follow-up visit and included the following: Autorefraction (Right group, Retinomax K-plus 3, Tokyo, Japan) during both non-cycloplegia and cycloplegia (cycloplegia was achieved by applying cyclopentolate 1% eye drops twice in each eye (Minims Cyclopentolate Hydrochloride 1%, Bausch & Lomb Nordic AB, Stockholm, Sweden) in-between a five-minute wait and then a thirty-minute wait to ensure cycloplegia). SER was determined as half of the cylindrical refraction combined with the spherical refraction. Push-plus subjective refraction was determined by using autorefraction and the current prescription as starting points. Optical biometry (IOLMaster 700, Carl Zeiss AG, Oberkochen, Germany) was used to measure AL, ACD, central corneal thickness (CCT) and lens thickness. Scheimpflug imaging (Oculus GmbH, Pentacam HR System, Wetzlar, Germany) was used to determine iridocorneal angle. An HOTV chart (Precision Vision, La Salle, IL, USA) was used to determine the best-corrected visual acuity (BCVA) at distance and near (4 m and 40 cm). The amplitude of accommodation was determined via a Royal Air Force near point ruler, while the participant used best-corrected spectacles for distance vision. Pupillometry (DP-2000 Pupillometer, NeurOptics, Irvine, CA, USA) was used to measure pupil diameter under both photopic (300 lux) and mesopic (4 lux) conditions, as the mean of five measurements. Swept source optical coherence tomography (OCT, Topcon Europe Medical BV, Capelle aan den Ijssel, The Netherlands) was used before pupil dilation to measure sub-foveal choroidal thickness and calculated via the built-in software (IMAGEnet 6, Topcon Europe Medical BV, Zoetermeer, The Netherlands). Tonometer (iCare Finland Oy, iCare, Vantaa, Finland) was used to determine intra-ocular pressure (IOP), and the mean of five measurements was used. Side effects were assessed and questioned at each visit by the examiners to determine whether participants suffered from ocular, periocular, visual, or anticholinergic side effects. Participants were asked whether they experienced blurred vision near or far, or photophobia. Participants were also questioned whether they experienced ocular side effects, including itching, eye redness/irritation, changes in tear production, pain and any allergic reactions. Side effects related to the eye surroundings that also were assessed included edema, skin changes, or dryness. Systemic anticholinergic side effects assessed were dryness of the skin, throat or mouth, symptoms from the gastrointestinal tract, facial redness, tachycardia, or urinary retention.

### 2.7. Statistical Analysis

Linear mixed models were structured with research facility and treatment as fixed effects using the R statistical software version 4.2.0 (R Program for Statistical Computing, Vienna, Austria) [32] with the LMMstar statistical package [33]. Individual eyes were included in the analysis as separate samples of measurement and a binary variable indicating right or left eye was nested in participant ID and added as a random effect to account for the correlation between eyes in the same individual. An unstructured covariance pattern was used to account for the potential variance heterogeneity over time, correlation between measurements from the same facility and correlation in the repeated measurements. Baseline values were assumed to be equal between intervention groups for the linear mixed modeling. A separate, second model was constructed to examine whether adjusting for iris color affected the effect estimates of the intervention. Data were analyzed with the intention-to-treat method. Multiple comparisons adjustment using the False Discovery Rate (FDR) [34] was used for *p*-values of secondary outcomes. Effect estimates with an adjusted *p*-value (adj-*p*) < 0.05 were considered statistically significant. Primary outcomes determining treatment efficacy (AL and SER) were excluded from the FDR adjustment.

## 3. Results

In total, 124 candidates were screened for participation (Figure 1). Sixteen failed inclusion criteria, six declined to participate following screening, two failed to comply with eye drop regimen tested by lubricating eye drops prior to randomization and three failed to comply with examinations. Finally, 97 participants with a mean age of 9.4 years (range 6–12) were included in the study and randomized to one of the interventions. Baseline mean SER and AL across all groups was −2.99 D (SD 1.27) and 24.6 mm (SD 0.84), respectively. Fifty-seven (57%) of participants were females, 59% had blue iris pigmentation, 31% had brown iris pigmentation and 10% had green iris pigmentation. Ethnically, 82 (85%) of participants were Caucasian, nine (10%) were mixed, three (3%) were Middle Eastern and two (2%) were of Asian origin. Five (5%) participants were excluded before the two-year visit: One participant wanted to try other another myopia intervention. Two participants withdrew consent. One participant was excluded after the twelve-month visit because of relocation to another country. One participant was lost to follow-up after completing the 18-month visit. Ultimately, 92 (95%) participants completed the two-year visit, of which 32 (35%) were in the 0.1% loading dose group, 32 (35%) were in the 0.01% group and 28 (30%) were in the placebo group (Figure 1). The drops were used at least six times per week (75%+ compliance rate) by all but one participant.

### 3.1. Axial Length and Spherical Equivalent Refraction Changes after Two Years

After two years of treatment, the mean AL was 25.09 mm (95% confidence interval (CI): 24.89; 25.30) in the 0.1% loading dose group and 25.07 mm (95% CI: 24.86; 25.27) in the 0.01% group compared to 25.17 mm (95% CI: 24.97; 25.38) in the placebo group. The mean AL progression from baseline, compared to placebo, was 0.08 mm less (95% CI: −0.01; 0.17) in the 0.1% loading dose group and 0.10 mm less (95% CI: 0.01; 0.19) in the 0.01% group (Figure 2, Table 2). The effect was statistically significant compared to placebo in the 0.01% group (*p*-value = 0.02) but not in the 0.1% loading dose group (*p*-value = 0.08). Treatment effect was comparable for brown and blue irides when adjusting for an iris and treatment interaction in the second model (adj-*p* = 0.37 and adj-*p* = 0.39, for brown vs. blue irides in the 0.1% loading dose and 0.01% groups, respectively). Only the effects of brown vs. blue irides on AL were compared because the number of participants with green eyes was too small.

Mean SER was −4.05 diopters (95% CI: −4.40; −3.70) in the 0.1% loading dose group and −3.91 diopters (95% CI: −4.26; −3.55) in the 0.01% group at two years follow-up compared to −4.17 diopters (95% CI: −4.52; −3.81) in the placebo group. Myopia progression was reduced by 0.12 diopters (95% CI: −0.10; 0.33) in the 0.1% loading dose group and 0.26 diopters (95% CI: 0.04; 0.48) in the 0.01% group compared to placebo, following two years of treatment (Figure 3). The effect was statistically significant compared to placebo in the 0.01% group (*p*-value = 0.02) but not in the 0.1% loading dose group (*p*-value = 0.30).

### 3.2. Anterior Chamber Depth, Corneal Thickness and Sub-Foveal Choroidal Thickness Changes after Two Years

The ACD, CCT, iridocorneal angle, lens thickness and sub-foveal choroidal thickness in all groups were comparable at the two-year visit (Appendix A).

### 3.3. Side Effects after Two Years

Mean IOP increased by 1.9 mmHg (95% CI: 0.8; 3.0, adj-*p* = 0.01) from baseline to the two-year visit in the placebo group, but mean IOP remained within normal limits and comparable between all groups at all visits (Table 2). Distance and near BCVA, amplitude of accommodation and mesopic and photopic pupil diameter were similar to baseline and comparable between all groups at the two-year visit.

A total of 17 adverse events and reactions (AE/AR) were found at the 18- and 24-month visits (Table 3). The most frequently reported AE/AR was eye redness/irritation (N = 5) (for a total of adverse events during the study, see Appendix A). As part of the study design, participants could be provided with near-vision add or photochromatic spectacles free of charge but no participants requested these. There were no serious adverse events or reactions reported during the second year of intervention.

## 4. Discussion

We analyzed the two-year effect and safety of 0.01% and 0.1% six-month loading doses followed by 0.01% atropine eye drops in reducing myopia progression in Danish children. We found a moderate reduction in myopia progression in both intervention groups, but the effect was only statistically significantly different from placebo in the 0.01% group.

We found a statistically significant two-year 0.10 mm reduction in axial elongation in our 0.01% group compared to placebo. When comparing the two-year overall progression in our placebo group and 0.01% (0.57 mm and 0.47 mm, respectively), it translates to a moderate treatment effect of an 18% reduction in AL progression in the 0.01% group compared to placebo, which is clinically relevant and is in line with the borderline significant one-year treatment effect we found in the 0.01% group in our previous interim analysis [24]. This is also comparable to the 19% two-year effect Zadnik et. al., 2023 found in their 0.01% group compared to placebo [27]. The MOSAIC study also found a significant 18% two-year reduction in AL progression of 0.01% atropine compared to placebo [25]. This is in contrast to Repka et al., 2023 who found no discernible effect of 0.01% compared to placebo [26]. All these studies were performed in North American/European settings, with large percentages of White children of comparable age groups, generally finding a significant effect of 0.01% low-dose atropine. Repka et al.’s contrasting findings could be due to the overweight Asian individuals in their 0.01% group compared to their placebo group (children of Asian ethnicity might have less benefit of 0.01% outside of an Asian setting [29]), their more ethnically heterogenous cohort (Table 1), or the inclusion of fewer participants with a moderate axial length (24 to 25 mm) at baseline in their 0.01% group compared to their placebo group, implying fewer future fast progressors in their 0.01% group and thereby, indirectly, a lesser treatment effect. Low-dose atropine seems to be a clinically efficacious treatment in populations outside of Asia.

Interestingly, our 0.01% group had performed better at two-year follow-up compared to the 0.1% loading dose group. Low-dose atropine treatment in higher concentrations has been associated with a hyperopic shift [35] and with a “rebound-effect” upon treatment cessation [18], which one study found to be most prominent during the first eight months after treatment cessation [36]. The ATOM 2 study speculated that 0.01% was superior to 0.1% in their study because of 0.01%’s higher efficacy in the second year and a subsequent lesser rebound compared to 0.1% [18]. The aim of our trial was to examine whether a 0.1% loading dose given in one group for 6 months followed by 18 months of 0.01% would increase efficacy and reduce a subsequent potential rebound effect. Our trial is still ongoing, and the final effect of a loading dose will first be apparent after the third-year washout period, but the reduction in effect in the 0.1% loading dose group could be seen as a rebound effect upon dosage reduction.

A possible explanation for the varying efficacy and side effects of low-dose atropine in different populations could, in part, be due to variation in iris pigmentation [21], perhaps via increased absorption by melanin pigment in brown irides leaving less drug available to bind to specific muscarinic receptors [37]. In our study, we observed no discernible difference in treatment effect when adjusting for iris pigmentation, although our sample size for this comparison was rather small. This observation is supported by German et al., 1999 who concluded that while increasing atropine dosage increased both binding to specific muscarinic receptors and non-specific binding to melanin pigment, the affinity for melanin pigment was much lower than for muscarinic receptors [37]. It seems that iris pigmentation does not significantly affect low-dose atropine’s efficacy in Danish children.

We observed a significant increase in mean IOP (~2 mmHg, at two-year vs. baseline) at the two-year visit in all groups, but no difference between groups. In other words, the IOP increased but not as a consequence of atropine treatment. The IOP is known to transiently increase with accommodation in myopes and non-myopes [38]. To prevent this, our IOP measurements were performed while participants were focused on a far point. Likewise, central corneal thickness is known to affect IOP measurements, but remained unchanged from baseline in all groups during the two-year period. We performed a post hoc analysis that showed this IOP increase was not associated with axial elongation (adj-*p* = 0.49), but rather time in the study (i.e., increasing age). An age-related IOP increase during childhood in healthy children has been reported [39,40,41], but it is typically reported to occur at an earlier age [41] and to be of a more modest degree [39]. This comparable larger childhood IOP increase in myopic children, albeit still within normal limits, could raise concerns about the future increased risk of glaucoma in this population.

A strength of this randomized clinical trial was its double-masked and placebo-controlled setup. A limitation was that this analysis was based on our interim two-year analysis. Final effects will first be apparent after the washout year.

## 5. Conclusions

In conclusion, we observed a statistically significant moderate reduction in myopia progression following two years of low-dose atropine treatment in Danish children. We observed a better treatment effect of 0.01% atropine compared to the 0.1% loading dose group, and the effect was only statistically significant in the 0.01% group at the two-year visit. Low-dose atropine is safe and has few adverse side effects; however, so far, the optimal dosing in Danish children remains to be established.

## Figures and Tables

**Figure 1 jpm-14-00175-f001:**
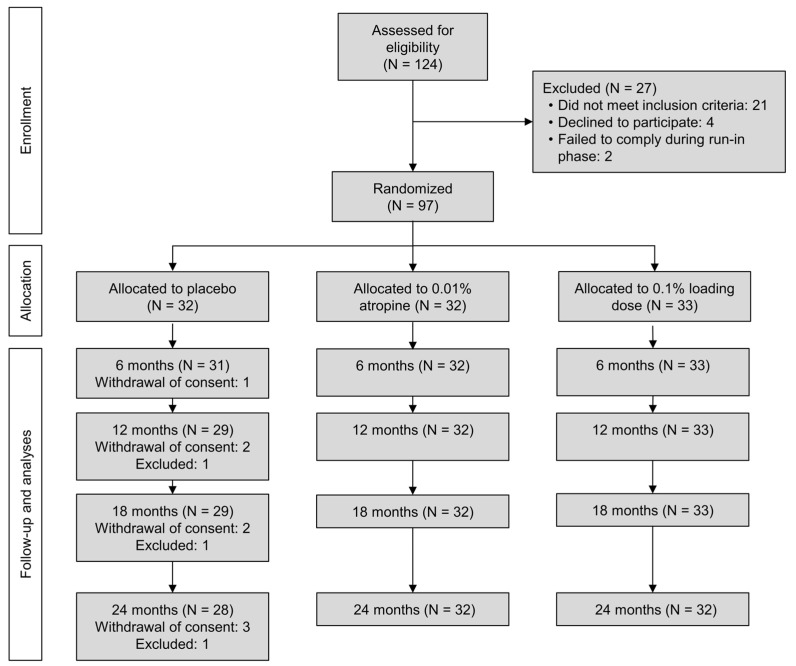
Flow-chart diagram (Consolidated Standards of Reporting Trials, CONSORT) of the study. Abbreviations: 0.1% loading dose, group that received 0.1% for the first six months followed by 0.01% for the following 18 months; 0.01%, group that received 0.01% for two years; N, number.

**Figure 2 jpm-14-00175-f002:**
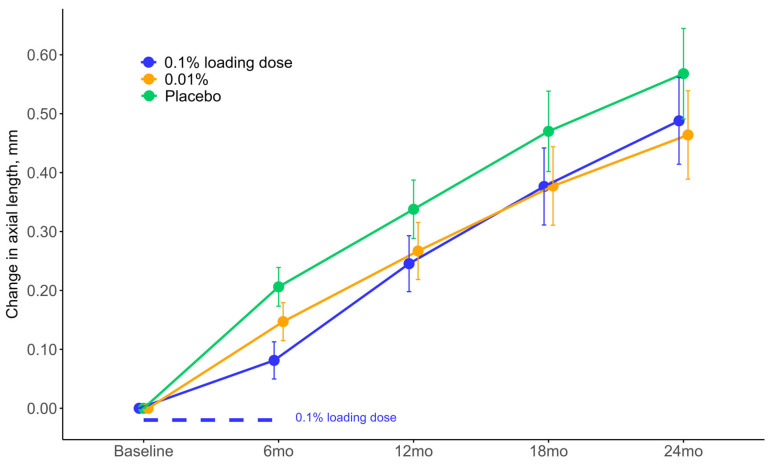
Changes in axial length during treatment with low-dose atropine or placebo in Danish myopic children. Mean change in axial length is estimated using linear mixed modeling by follow-up visit for the intervention groups. Error bars show the means’ 95% CI. Abbreviations: 0.1% loading dose, group that received 0.1% for six months and then 0.01% for 18 months; 0.01%, group that received 0.01% for two years; mm, millimeters; mo, month; placebo, group that received placebo eye drops for two years.

**Figure 3 jpm-14-00175-f003:**
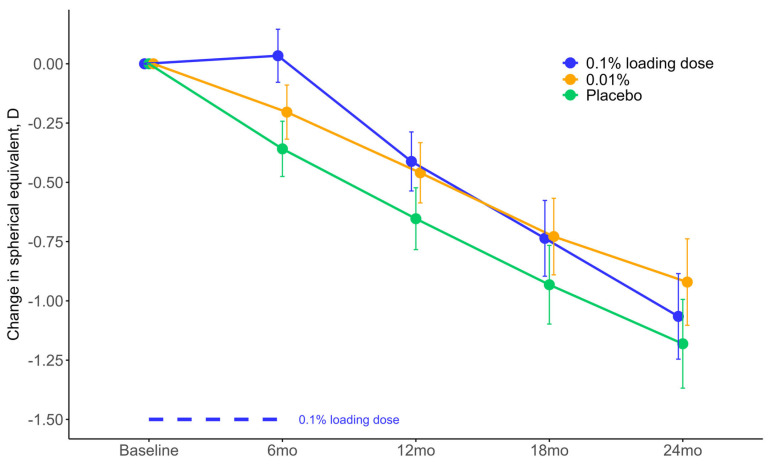
Changes in spherical equivalent refraction during treatment with low-dose atropine or placebo in Danish myopic children. Mean change in spherical equivalent refraction is estimated using linear mixed modeling by follow-up visit for the intervention groups. Error bars show the means’ 95% CI. Abbreviations: 0.1% loading dose, group that applied 0.1% for six months and then 0.01% for 18 months; 0.01%, group that applied 0.01% for two years; D, diopters; mo, month; placebo, group that applied placebo eye drops for two years.

**Table 1 jpm-14-00175-t001:** Comparison of 0.01% atropine efficacy on axial length elongation in recent studies with cohorts containing White participants.

Study	Published Year	Dose and Intervention Period	Age at Baseline, Mean (SD)	Axial Length at Baseline, Mean (SD)	% White Ethnicity	AL Change in mm Compared to Placebo	*p*-Value
Repka et al. [26]	2023	0.01% for 24 months	All *: 10.1 years (1.8)	All *: 24.4 mm (0.8)	46%	−0.002 (95% CI: −0.11; 0.10)	Not reported
Zadnik et al. [27] (CHAMP-study)	2023	0.01% for 24 months	Placebo: 8.8 years (1.8) 0.01%: 9.0 years (2.1)	Placebo: 24.33 mm (0.84) 0.01%: 24.37 mm (0.81)	56.4%	−0.12 (95% CI: −0.06; −0.18)	<0.01
Loughman et al. [25] (MOSAIC-study)	2023	0.01% for 24 months	Placebo: 11.78 years (2.17) 0.01%: 11.84 years (2.47)	Placebo: 24.93 mm (1.09) 0.01%: 24.85 mm (1.02)	80.8%	−0.07 (95% CI: −0.01; −0.13)	0.009
Lee et al. [28] (WA-ATOM-study)	2022	0.01% for 24 months	Placebo: 12.2 years (SD: 2.5) 0.01%: 11.2 years (SD: 2.7)	Placebo: 24.7 mm (IQR): 24.4–25.4 0.01%: 24.6 mm (IQR: 24.2–25.2)	50%	−0.05 (95% CI: 0.01; −0.11)	0.1
Hansen et al. [24] (APP-study)	2023	0.01% for 12 months	All *: 9.4 years (1.7)	All *: 24.60 mm (0.84)	85%	−0.07 (95% CI: 0.00; −0.15)	0.16 (FDR adjusted)

* Mean of all groups combined. Abbreviations: CI, confidence interval; FDR, False Discovery Rate; IQR, interquartile range; SD, standard deviation.

**Table 2 jpm-14-00175-t002:** Group treatment effect estimates for ocular parameters from linear mixed modeling.

	Group	Placebo	0.1% Loading Dose ^a^	0.01% ^b^
Time Point				
AL, mm				
Baseline		24.60 (24.42; 24.78)		
18 mo		25.08 (24.87; 25.28)	−0.09 (−0.17; −0.02)	−0.09 (−0.17; −0.01)
24 mo		25.17 (24.97; 25.38)	−0.08 (−0.17; 0.01)	−0.10 (−0.19; −0.01)
24 mo *p*-value			0.08	0.02 *
SER, diopters				
Baseline		−2.99 (−3.26; −2.71)		
18 mo		−3.92 (−4.26; −3.57)	0.20 (0.01; 0.39)	0.20 (0.01; 0.40)
24 mo		−4.17 (−4.52; −3.81)	0.12 (−0.10; 0.33)	0.26 (0.04; 0.48)
24 mo *p*-value			0.30	0.02 *
IOP, mmHg				
Baseline		15.8 (15.1; 16.5)		
18 mo		17.7 (16.9; 18.6)	0.3 (−0.7; 1.3)	0.09 (−0.9; 1.1)
24 mo		17.7 (16.7; 18.6)	0.2 (−0.9; 1.3)	−0.7 (−1.8; 0.5)
24 mo adjusted-*p*			0.87	0.67
Distance BCVA, LogMAR				
Baseline		−0.10 (−0.12; −0.09)		
18 mo		−0.13 (−0.15; −0.11)	0.00 (−0.02; 0.02)	0.01 (−0.01; 0.04)
24 mo		−0.12 (−0.14; −0.10)	0.00 (−0.02; 0.02)	0.00 (−0.02; 0.02)
24 mo adjusted-*p*			0.87	0.97
Near BCVA, LogMAR				
Baseline		−0.07 (−0.09; −0.05)		
18 mo		−0.09 (−0.11; −0.06)	0.00 (−0.02; 0.03)	0.01 (−0.02; 0.04)
24 mo		−0.08 (−0.11; −0.06)	0.00 (−0.02; 0.03)	0.01 (−0.02; 0.04)
24 mo adjusted-*p*			0.82	0.84
Accommodation amplitude, diopters				
Baseline		16.4 (15.6; 17.2)		
18 mo		16.7 (15.8; 17.7)	−0.4 (−1.5; 0.6)	0.2 (−0.9; 1.3)
24 mo		16.8 (15.8; 17.8)	−0.5 (−1.5; 0.6)	−0.8 (−1.8; 0.3)
24 mo adjusted-*p*			0.76	0.57
Mesopic pupil diameter, mm				
Baseline		4.28 (4.08; 4.49)		
18 mo		4.33 (4.07; 4.60)	0.12 (−0.18; 0.42)	−0.03 (−0.33; 0.27)
24 mo		4.35 (4.10; 4.60)	−0.04 (−0.32; 0.23)	0.09 (−0.18; 0.37)
24 mo adjusted-*p*			0.87	0.80
Photopic pupil diameter, mm				
Baseline		2.80 (2.67; 2.94)		
18 mo		2.74 (2.49; 2.99)	0.06 (−0.14; 0.26)	−0.06 (−0.26; 0.13)
24 mo		2.77 (2.62; 2.91)	0.03 (−0.14; 0.19)	0.06 (−0.10; 0.23)
24 mo adjusted-*p*			0.87	0.77

Effect estimates are presented as total for the placebo group and as differences from the placebo group at the different time points for the 0.1% loading dose and 0.01% intervention groups. Significance levels for exploratory secondary outcomes were reported as adj-*p*, while primary outcomes determining treatment efficacy were excluded from FDR-adjustment and reported as *p*. Abbreviations: AL, axial length; BCVA, best-corrected visual acuity; mo, months; *p*, *p*-value; adjusted-*p*, *p*-value adjusted by False Discovery Rate; SER, spherical equivalent refraction. ^a^ Change compared to placebo in the 0.1% loading dose group at that time point. ^b^ Change compared to placebo in the 0.01% group at that time point. * Statistically significant if below *p*-value or adjusted-*p* cut-off of 0.05.

**Table 3 jpm-14-00175-t003:** Adverse events for the second year of intervention.

Group	Event	12 mo	18 mo	24 mo
0.1% loading dose	Total events, N/total N (%)	2/33 (6%)	2/32 (6%)	5/32 (16%)
	Eye redness/irritation, N/total N (%)	1/33 (3%)	0/32 (0%)	1/32 (3%)
	Photophobia, N/total N (%)	0/33 (0%)	0/32 (0%)	1/32 (3%)
	Blurred near vision, N/total N (%)	0/33 (0%)	0/32 (0%)	1/32 (3%)
	Blurred distance vision, N/total N (%)	0/33 (0%)	0/32 (0%)	0/32 (0%)
	Other, N/total N (%)	1/33 (3%)	2/32 (6%)	2/32 (6%)
	Dilated pupils, N/total N (%)	0/33 (0%)	0/32 (0%)	0/32 (0%)
0.01%	Total events, N/total N (%)	1/32 (3%)	4/32 (13%)	1/32 (3%)
	Eye redness/irritation, N/total N (%)	0/32 (0%)	2/32 (6%)	0/32 (0%)
	Photophobia, N/total N (%)	0/32 (0%)	1/32 (3%)	0/32 (0%)
	Blurred near vision, N/total N (%)	0/32 (0%)	0/32 (0%)	0/32 (0%)
	Blurred distance vision, N/total N (%)	0/32 (0%)	0/32 (0%)	0/32 (0%)
	Other, N/total N (%)	1/32 (3%)	1/32 (3%)	1/32 (3%)
	Dilated pupils, N/total N (%)	0/32 (0%)	0/32 (0%)	0/32 (0%)
Placebo	Total events, N/total N (%)	2/29 (6%)	2/29 (7%)	3/28 (11%)
	Eye redness/irritation, N/total N (%)	0/29 (0%)	1/29 (3%)	1/28 (4%)
	Photophobia, N/total N (%)	0/29 (0%)	0/29 (0%)	1/28 (4%)
	Blurred near vision, N/total N (%)	1/29 (3%)	0/29 (0%)	0/28 (0%)
	Blurred distance vision, N/total N (%)	0/29 (0%)	0/29 (0%)	0/28 (0%)
	Other, N/total N (%)	1/29 (3%)	1/29 (3%)	1/28 (4%)
	Dilated pupils, N/total N (%)	0/29 (0%)	0/29 (0%)	0/28 (0%)

“Total events” indicate the total number of participants who experienced at least one adverse event. Abbreviations: N, number of participants; mo, month.

## Data Availability

The anonymized datasets from the current study are available from the corresponding author upon reasonable request.

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
