# Peer review of "Two-Year Results of 0.01% Atropine Eye Drops and 0.1% Loading Dose for Myopia Progression Reduction in Danish Children: A Placebo-Controlled, Randomized Clinical Trial"

_jpm, 2024, doi:10.3390/jpm14020175_

Round 1
Reviewer 1 Report
Comments and Suggestions for Authors
This study is a much needed addition to the issue of atropine in myopia treatment, and it brings valuable data to practitioners worldwide. The study methodology is complex, rigorous and suitable for strong conclusions. The work is well researched and comprehensive in its approach.
Author Response
Dear Reviewer
Thank you for reviewing our manuscript and for the great response. We are pleased that you liked the paper.
Best regards,
Niklas Cyril Hansen
Reviewer 2 Report
Comments and Suggestions for Authors
The article presents the results of a placebo-controlled, double-masked,randomized clinical trial, which analysed over a period of two years the effect and safety of 0,01% and 0,1% six months loading dose followed by 0,01% atropine eye drops in reducing myopia progression in Danish children. Primary outcomes were axial length and spherical equivalent refraction. Secondary outcomes included adverse reactions and other ocular biometrical measures. The aim of the trial was to examine whether a 0,1% loading dose given in one group for six months followed by 18 months of 0.01% would increase the efficacy and reduce the subsequent rebound effect. The conclusion was; the 0,01% group had performed better at two year follow-up compared to the 0,1% loading dose group.The article is fluently written, with details regarding the studied groups , the investigations and the statistical analysis methods used. The conclusions are well argued and provide useful data in the treatment of myopia progression in children.
Author Response
Dear Reviewer
Thank you for reviewing our manuscript and for the the great reception of our manuscript. We are pleased that you liked the paper.
Best regards,
Niklas Cyril Hansen
Reviewer 3 Report
Comments and Suggestions for Authors
L46 Multifocal spectacle lenses should be changed to multifocal Segments spectacle lenses .
L51 Drops should be changed to be drop.
L99-100 What is the basis for such including children?
L265-267 Table1 and Table2 does not show ACD, CCT, iridocorneal angle, lens thickness and sub-foveal choroidal thickness datas, please supplement these datas.
L274-277 What were the managements and outcomes of these side effects?
L286 Such a small sample size does not support this conclusion.If you insist, please give enough evidence.
L311-371 ATOM2 study does not explain why the clinical effect was worse in the 0.1% loading dose group than in the 0.01% group without a washout-period.

Comments on the Quality of English LanguageThere are some minor flaws in English writing, but the whole artical is very good!
翻译来自彩云小译Author Response
Dear Reviewer
Thank you for the thorough review of our manuscript. We have attached a cover-letter/point-by-point response to your points here and hope they adequately address your concerns.
Best regards,
Niklas
